# DNA Methylation in Pituitary Adenomas: A Scoping Review

**DOI:** 10.3390/ijms26020531

**Published:** 2025-01-10

**Authors:** Morten Winkler Møller, Mathias Just Nortvig, Mikkel Schou Andersen, Frantz Rom Poulsen

**Affiliations:** 1Department of Neurosurgery, Odense University Hospital, DK-5000 Odense, Denmark; mathias.just.nortvig@rsyd.dk (M.J.N.); mikkel.c.schou.andersen@rsyd.dk (M.S.A.); frantz.r.poulsen@rsyd.dk (F.R.P.); 2Department of Clinical Research, University of Southern Denmark, DK-5000 Odense, Denmark; 3BRIDGE (Brain Research-Inter Disciplinary Guided Excellence), University of Southern Denmark, DK-5000 Odense, Denmark

**Keywords:** pituitary adenomas, DNA methylation, epigenetics

## Abstract

Pituitary adenomas are a diverse group of neoplasms with variable clinical behavior. Despite advances in genetic analysis, understanding the role of epigenetic modifications, particularly DNA methylation, remains an area under investigation. This scoping review aimed to update and synthesize the current body of literature on DNA methylation in pituitary adenomas, focusing on methodological advancements and clinical correlations. A systematic search conducted across multiple databases, including Embase, Scopus, MEDLINE, and CENTRAL, identified 107 eligible studies. Early methods, such as methylation-restricted digestion and methylation-specific PCR (MSP), have evolved into more comprehensive approaches, such as chip-based DNA methylation analysis. Key findings suggest that genes like POMC, SOCS-1, and RASSF1A show a significant association between methylation and clinical behavior. However, methylation patterns alone are insufficient to fully explain tumorigenesis. Emerging data suggest that DNA methylation might serve as a prognostic marker for invasive growth and recurrence, but further longitudinal studies are needed. This review highlights the need for future research to explore the methylome more thoroughly and to better define the clinical impact of epigenetic modifications in pituitary adenomas.

## 1. Introduction

In 2017, the World Health Organization (WHO) updated the classification of pituitary adenomas with the addition of transcription factors SF1, PIT1, and TPIT to better delineate adenohypophyseal cell lineages [1]. Transcription factor staining thus divides pituitary adenomas into three distinct subgroups, where SF1 is expressed in gonadotroph adenomas, PIT1 is expressed in GH- (somatotroph), PRL-, and TSH-producing adenomas, and TPIT is expressed in corticotroph adenomas [2]. Pituitary adenomas are further classified as either functioning pituitary adenomas (FPA) that secrete hormones to peripheral blood or nonfunctioning pituitary adenomas (NFPA) that do not secrete hormones [3]. The classification of pituitary adenomas based solely on transcription factors and hormone secretion does not fully explain their clinical presentation and tumor differentiation [4]. To address this, the Pituitary Society introduced a nine-tiered prognostic classification during the Pituitary Adenoma Nomenclature 3 (PANOMEN 3) clinical classification workshop. This classification combines preoperative tumor size with tumor phenotype and clinical behavior, such as invasion and mass effect (11). Research on pituitary tumorigenesis has identified a few hereditary mutations, such as *MENIN* [5] and *AIP* [6], that predispose to pituitary adenomas. Somatic mutations in *GNAS* (associated with somatotroph adenomas) [7] and *USP8* (corticotroph adenomas) [8] have also been implicated in pituitary adenomas. Exon sequencing has otherwise revealed little additional genetic association with pituitary adenomas [9].

The limited insights from genetic studies have changed the narrative towards epigenetic alterations [10,11], such as DNA methylation [12] and histone modification [13], that may contribute to tumorigenesis, invasive growth, and disease progression. Advances in transcriptomic analysis have strengthened the classification of pituitary adenomas [14], and DNA methylation has shown potential for further improvement in their subgrouping [15].

DNA methylation is a key regulator of gene expression [16], and is mediated by DNA methyltransferases (DNMTs) that transfer a methyl group to the C5 position of cytosine nucleotides. Such 5-methylcytosines that are positioned upstream to a guanine nucleotide in DNA sequences are referred to as CpG sites [17]. The accumulation of CpG sites often results in transcriptional silencing, particularly when clustered as CpG islands [18]. As shown in Figure 1, DNA methylation inhibits the binding of transcription factors causing no or lower generation of mRNA, leading to a lower expression of the gene products generated from the specific methylated DNA strand. These products are either secreted to the extracellular space or maintained within the intracellular space regulating signaling pathways.

The role of DNA methylation in pituitary adenomas has been investigated for several decades, aiming to address the diagnostic challenges that arise due to the diverse clinical and immunohistochemical profiles of these tumors [19,20]. DNA methylation has been correlated with specific preoperative conditions, like invasive behavior [21] and postoperative recurrence [22]. The current knowledge about the value of DNA methylation and its association with clinical outcomes in pituitary adenomas is limited, as the molecular signatures are not incorporated in the guidelines when risk assessing pituitary adenomas [23]. Epigenetic mechanisms, including DNA methylation, are increasingly seen as fundamental to understanding tumor behavior and heterogeneity [24,25], including pituitary adenomas [26]. The latest systematic review on DNA methylation in pituitary adenomas was conducted in 2013 [16], underscoring the need for an updated synthesis.

The aim of this scoping review was to update and summarize the literature on advancements in DNA methylation in pituitary adenomas, highlighting trends and knowledge gaps for future research directions.

### 1.1. Three Main Methods to Perform DNA Methylation

To date, three methods have primarily been used to assess DNA methylation in pituitary adenomas, each offering unique insights into methylation patterns. Alongside Whole Genome Bisulfite Sequencing (WGBS), which profiles over 28 million CpG sites [27] and has never have been performed on pituitary adenomas, early studies relied on a labor-intensive digestion-based approach. This has evolved into methylation-specific PCR (MSP) that targets specific methylated regions and, most recently, a high-throughput chip-based methylation analysis covering over 200,000 sites for more comprehensive, genome-wide analysis.

#### 1.1.1. Methylation-Restricted Digestion

This earliest approach, methylation-restricted digestion, used restriction enzymes to fragment DNA at specific methylation-sensitive sites [19]. Following digestion, the DNA could undergo PCR amplification for targeted regions [28] or be separated using agarose gel electrophoresis. DNA fragments were then transferred to a nitrocellulose membrane and probed for specific methylated regions through hybridization. Although this technique was pivotal in early methylation studies, it was limited by the specificity of restriction enzymes and the available knowledge of methylation sites.

#### 1.1.2. Methylation-Specific Polymerase Chain Reaction (MSP)

This approach uses bisulfite conversion, which selectively converts unmethylated cytosine residues to uracil while methylated cytosines remain unchanged. This conversion allows the differentiation of methylated from unmethylated DNA, as unmethylated cytosines are ultimately converted to thymine during PCR [29]. Commercial kits streamline the bisulfite conversion process, making MSP a widely accessible technique. The bisulfite-treated DNA is then amplified using two sets of primers that target either methylated or unmethylated CpG sites [30]. MSP is highly specific, but is constrained to predefined regions, limiting its use for broader genomic exploration.

#### 1.1.3. Chip-Based DNA Methylation Analysis

Chip-based DNA methylation analysis, or DNA methylation arrays, is a high-throughput method that enables the genome-wide profiling of methylation across thousands of CpG sites [31]. After bisulfite conversion, the DNA is fragmented and hybridized to microarray probes specific to CpG sites throughout the genome. The probes selectively bind based on methylation status, with fluorescent labels indicating methylation at each site. Data from the array are processed to generate comprehensive methylation profiles, revealing differentially methylated positions (DMPs) and regions (DMRs) linked to specific phenotypes or disease states. This method provides a global perspective on DNA methylation, enhancing our understanding of its role in tumor behavior and progression. 

## 2. Results

### 2.1. Literature Screening and Eligibility

A comprehensive systematic literature search across Embase, Scopus, MEDLINE, and CENTRAL yielded 1140 abstracts. An additional two studies were identified through forward and backward citation searches. After duplicate removal, 439 abstracts were excluded, with 434 duplicates identified through Covidence systematic review web-based collaboration software platform (Veritas Health Innovation, Melbourne, Australia), and 5 were removed manually. Of the remaining 703 studies, 508 were excluded based on their title and abstract review. This left 195 studies eligible for full-text review, all of which were successfully retrieved. Three older studies that were inaccessible through standard databases were obtained with the support of a healthcare librarian at the University of Southern Denmark.

Following full-text screening, 88 studies were excluded due to irrelevant comparators (e.g., studies focusing on histone modifications without DNA methylation, immunohistochemical staining for *MGMT*, or transcription analysis without methylation data); inappropriate study design (e.g., conference abstracts, studies on chemotherapeutic effects, or narrative reviews not identified in the abstract stage); or unsuitable study population (e.g., pediatric studies or studies on intracranial tumors other than pituitary adenomas).

In total, 107 studies met the eligibility criteria and were included in this scoping review (see Figure 2).

### 2.2. Study Characteristics

The studies included in this review are detailed in Appendix A and Table 1. While 8 studies used methylation-restricted digestion (Appendix A), 59 used methylation-specific PCR (MSP) (Appendix A), and 31 used chip-based microarray analyses (Table 1). Each study examined DNA methylation in pituitary adenoma tissue, exploring its association with various genes, pathways, and gene expression patterns relevant to tumor behavior. Additional papers using alternative approaches (nine studies) are summarized in Appendix A.

#### 2.2.1. Methylation-Restricted Digestion

Due to their labor-intensive nature, methylation-restricted digestion studies typically involved small sample sizes, with a maximum of 30 samples [32]. Genes analyzed through this method included *GH* [19,33], *p16* [28,34], the *RB1* promoter region [32], the *POMC* promoter region [30], *1A DMR* [35], and the *MGMT* promoter region [36]. These studies commonly demonstrated correlations between methylation status and gene expression. The earliest investigation of DNA methylation in pituitary adenomas, conducted in 1988, examined the methylation of the *GH* gene in relation to hypersecretion in GH-positive adenomas, and suggested that hypomethylation might be linked to overexpression [33]. Although sample sizes gradually increased, later studies—including analyses of *MGMT* promoter methylation—did not establish this marker as predictive of temozolomide response in aggressive adenomas. Additional findings from studies using methylation-restricted digestion are presented in Appendix A.

#### 2.2.2. Methylation-Specific PCR

The advent of MSP allowed larger sample sizes and more precise assessments of methylation across specific genomic regions. The increased efficiency of MSP, aided by commercially available kits, enabled the detailed exploration of methylation patterns and correlations with gene expression data and clinical behavior. Strong correlations between methylation, gene expression, and clinical outcomes were found for genes involved in cell cycle regulation, apoptosis, and various signaling pathways such as *CDKN2A* [37,38], *DAP kinase gene* [25,39], *GADD45g* [40], *RASSF1A* [41], *RASSF3* [42], *Gal-3* [43], *SOCS-1* [44], *FGFR2-IIIb* [45], *GSTP1* [46,47], *NNAT* [48], *EFEMP1* [49], *sFRP4* [50,51,52], *p21* [53], *p27* [53], *WIF1* [52], *GIPR* [54,55], *DNMT1* [56], *DNMT3A* [56], *DNMT3B* [56], *LAMA2* [57], *TERT* [58,59,60], *GNAS* [61,62], and the *POMC* promoter [63].

Genes and pathways with a probable, but not statistically significant, association with clinical behavior included the *E-cadherin* gene [64], *C22orf3* [65], *MEG3* [26,66], *p15INK4b* [67], *RB1* [25,38,67,68], *CDH13* [69], *FGFR2* [70], *CDKN2C* [38], *RIZ1* [71], *SSTR5* [72], *Pttg1* [73], autosomal and X-linked genes [74], and the *CHST7* promoter [75].

Genes such as *p16* [76,77,78], *CDH1* [69,79], *p18INK4C* [80], *NDRG2* [81], *GADD45b* [82], *MEN1* [83], the *hTERT* promoter [84], and the *STAT3* promoter [85] showed no association with clinical differences between pituitary adenomas and normal tissue. Likewise, the *MGMT* promoter methylation [25,86,87,88,89,90,91,92] did not appear to be a reliable prognostic marker for responses to temozolomide in aggressive adenomas. Additional genes including *p16INK4a* (*CDKN2A*) [25,67,68], *CDK4* [67], *p14ARF* [25,38], *p73* [25], *THBS1* [25], *caspase 8* [25], and *TIMP-3* [25] were investigated, but no definitive conclusions were drawn regarding the impact of methylation on expression or clinical outcomes.

While the list of genes investigated is extensive, no single gene or pathway has been identified as the sole driver of tumor regrowth, hypersecretion, or invasiveness. More recent studies using MSP emphasize a shift towards examining broader molecular patterns, potentially contributing to tumor differentiation rather than single-pathway mechanisms [93]. Detailed findings are provided in Appendix A).

#### 2.2.3. Chip-Based DNA Methylation Analysis

The first genome-wide, methylation-based analyses of pituitary adenomas revealed multiple candidate genes and signaling pathways that enabled differentiation between pituitary adenomas and other sellar tumors, as well as between adenoma subtypes that had traditionally been distinguished using immunohistochemistry [14,15,94,95,96,97,98,99,100,101,102,103,104]. Genome-wide methylation patterns also highlight differences between clinically invasive and non-invasive tumors, thus identifying several genes that may contribute to invasiveness without a definitive characterization [11,105,106,107,108].

Several studies identified specific genes and pathways associated with invasive tumor behavior; for example, *ITPKB* and *CNKSR1* [109]. Additional genes—*PHYHD1*, *LTBR*, *MYBPHL*, *C22orf42*, *PRR5*, *ANKDD1A*, *RAB13*, *CAMKV*, *KIFC3*, *WNT4*, and *STAT6*—were implicated in the invasive behavior of non-functioning pituitary adenomas (NFPAs) [21]. *TERT* was examined, but no specific pathogenic role was identified in pituitary tumors [110]. Genes such as *STAT5A*, *RHOD*, *GALNT9*, *RASSF1*, *CDKN1A*, *TP73*, *STAT3*, *HMGA2*, *FAM163A*, *HIF3A*, and *PRSS8* were found to be hypermethylated in NFPAs [111]. Six genes (*FAM90A1*, *ETS2*, *STAT6*, *MYT1L*, *ING2*, and *KCNK1*) were identified as potential biomarkers for NFPA regrowth [112]. A study correlating methylation and gene expression identified several hub proteins associated with pituitary adenomas, including *DCC*, *DLG5*, *ETS2*, *FOXO1*, *HBP1*, *HMGA2*, *PCGF3*, *PSME4*, *RBPMS*, *RREB1*, *SMAD1*, *SOCS1*, *SOX2*, *YAP1*, and *ZFHX3* [24]. Myc-associated protein X binding sites were globally hypomethylated in growth hormone-secreting pituitary adenomas (GHPAs) compared to NFPAs, suggesting increased transcription factor binding accessibility as a potential therapeutic target for GHPAs [113]. Additionally, one study found no differences between GHPAs and NFPAs, but did find genes *C7orf50*, *GNG7*, and *BAHCC1* to be associated with postoperative progression [114]. Tumors from the posterior pituitary lobe showed only minor methylation differences [115].

A recent study proposed that DNA methylation patterns in circulating cell-free DNA may distinguish pituitary adenomas from other sellar pathologies, enabling non-invasive lesion characterization based on blood samples [116].

Comprehensive findings from these studies are summarized in Table 1.
ijms-26-00531-t001_Table 1Table 1Chip-based methylation analysis.Author/YearAimSample SizeKey FindingsDuong et al. [94]/2012Methylation profile of 27,578 CpG sites spanning more than 14,000 genes in each of the major pituitary adenoma subtypes7 GH-secreting tumors, 6 corticotrophinomas, 6 prolactinomas (PRL), and 13 non-functioning (NF) adenomasFirst and unbiased survey of the pituitary tumor epigenome across different adenoma subtypesLing et al. [95]/2014DNA methylation alterations between invasive and noninvasive PAs subtypes24 patients with surgically resected PAsDNA methylation analysis of key candidate genes may potentially complement histopathological classification systems for PA subtypesGu et al. [105]/2016DNA methylation differences between invasive and non-invasive non-functioning PAs12 adenomas were included in the discovery cohort; 7 adenomas were included in an independent cohortEpigenetic modification of key gene substrates might partially account for the invasion of non-functioning PAsKober et al. [109]/2018The role of aberrant methylation at particular loci for gene expression in PAs31 gonadotroph NFPAs, 2 NFPAs that were positive for gonadotropins (FSH, LH, a-subunit) and TSH, and 1 null-cell adenomaInvasive NFPAs showed invasiveness-related aberrant epigenetic upregulation of *ITPKB* and downregulation of *CNKSR1*Boresowicz [110]/2018Incidence of *TERT* abnormalities and to assess their role in telomere lengthening in PAsTissue samples from 101 patientsTelomerase abnormalities do not play any special role in pathogenesis of pituitary tumorsJohann et al. [96]/2018Characterize molecular alterations of sellar region ATRTs in adults as compared to pituitary adenomas47 pituitary adenomas were evaluatedSellar region ATRTs in adults form a clinically distinct entity with a different mutational spectrumSalomon et al. [97]/2018DNA methylation data were generated from the three major subtypes of pituitary adenomas48 patientsDNA methylation alterations play a major role in the disease etiologyKober et al. [111]/2019DNA methylation in the misregulation of gene expression in gonadotroph NFPAs32 patientsGenes with aberrant methylation in pituitary tumors—*STAT5A*, *RHOD*, *GALNT9*, *RASSF1*, *CDKN1A*, *TP73*, *STAT3* and *HMGA2*. *FAM163A*, *HIF3A*, and *PRSS8*—were hypermethylated in NFPAsCheng et al. [21]/2019Integrated analyses of paired whole-genome DNA methylation and gene expression in PARetrospectively enrolled 68 patientsMethylation and expression levels of *PHYHD1*, *LTBR*, *MYBPHL*, *C22orf42*, *PRR5*, *ANKDD1A*, *RAB13*, *CAMKV*, *KIFC3*, *WNT4*, and *STAT6* play a pivotal role in the invasive behavior of NFPANeou et al. [98]/2019A molecularly unbiased classification, further deciphering the pathways responsible for tumorigenesis from a single set of PitNETsThe methylome of 86 PitNETs of all typesIdentified three groups associated with tumor type and secretion; in particular, POU1F1/PIT1-lineage tumors showed global hypomethylationCheng et al. [112]/2020DNA methylation and expression parameters to evaluate the regrowth of NFPA71 patients diagnosed with NFPA6 of 13 genes (*FAM90A1*, *ETS2*, *STAT6*, *MYT1L*, *ING2*, and *KCNK1*) were considered potential biomarkers associated with the regrowth of NFPABoresowicz et al. [106]/2020CpGs located in miRNA genes that have differential methylation levels in gonadotroph PitNETs34 PitNETs and 5 samples of normal pituitaryEpigenetic regulation and changes in miRNA expression play a significant role in pathogenesis of PitNETsTaniguchi-Ponciano et al. [14]/2020Identify the cellular pathways involved in their tumorigenesis6 non-tumoral pituitaries and 42 PAsA divergent PA origin that segregates transcriptomically into three distinct clusters depending on the specific transcription factorsWei et al. [11]/2020Activated and inhibited pathways and related key genes in hpNFPAs versus NFPAsEight snap-frozen NFPA specimens (four hpNF-PAs and four NFPAs)The DNA methylation and gene expression patterns of two highly proliferative NFPAs occurring at young ages were noticeably distinct to those of six other NFPAsMosella et al. [15]/2021Identify, characterize, and validate methylation-based signatures that define PitNETs according to clinicopathological featuresDNA methylation data from PitNETs from three independent institutions and from our cohort at the Hermelin Brain Tumor Center (n = 23)Methylation signatures distinguished PitNETs by adenohypophyseal cell lineagesSchmid et al. [115]/2021Molecular differences between the three histologic types using DNA methylation analysis47 neoplasms of the posterior pituitary glandOnly subtle DNA methylation differences among tumors of the posterior pituitaryNadaf et al. [99]/2021Use generated data to gain insights into the initiation and development of PitBs by identifying pathways differentially alteredA total of 64 tumor and normal FFPE tissuesPitB samples formed a distinct cluster separate from the various pituitary adenoma subtypesHagel et al. [107]/202112 double PAs (DPA) with diverse hormone profiles were investigated regarding DNA methylation profile12 cases were identified among 3654 surgical specimens of adenomaGlobal DNA methylation profiling may yield additional information in lesions that appear as null-cell adenomas immunohistochemicallyAsuzu et al. [117]/2022A mechanism of tumorigenesis with therapeutic implications for CDThree tumors investigated for methylation (EPIC)There may exist histone modifications that contribute to the pathogenesis of wild-type CD that were not captured by DNA methylation approachesDottermusch et al. [118]/2022(Case report)The role of epigenomic analyses in the diagnostic workup of a challenging sellar lesion57-year-old maleExemplifies benefits and limitations of epigenomic analyses in molecular diagnostics of posterior pituitary neoplasmsHickman et al. [119]/2022(Case report)A functioning corticotroph tumor with admixed adrenocortical cells, providing novel methylation profiling data33-year-old maleThe methylation profile of this tumor was unique but was placed within the T-SNE plots adenohypophyseal entities, with the closest match being corticotroph tumorsGiuffrida et al. [114]/2022Correlate the methylation status of NFPAs and GH-omas with their epidemiological and clinicopathological features21 PA samples (11 GH-omas, 10 NFPAs)*C7orf50*, *GNG7*, and *BAHCC1* genes, which were found to be methylated in pituitary tumor biologyHallén et al. [108]/2022Whether DNA methylation pattern differs in NFPAs between patients with residual adenoma with postoperative progression28 tumors from the reintervention group and 21 tumors from the radiologically stable groupMethylation patterns associated with clinically significant tumor growth requiring reinterventionSilva-Júnior et al. [100]/2022Methylome and transcriptome analysis of the three major subtypes of surgically resectable PitNETs77 patients (46 NFPT and 31 functioning pituitary tumors)Methylome and transcriptome data resulted in three clusters that were associated with each other and with 2017 and 2022 WHO classificationsAydin et al. [24]/2022Evaluate the molecular profiling of NF-PitNETs at three biological levels34 NF-PitNET samples and 6 normal pituitary glandsProposed hub proteins, including *DCC*, *DLG5*, *ETS2*, *FOXO1*, *HBP1*, *HMGA2*, *PCGF3*, *PSME4*, *RBPMS*, *RREB1*, *SMAD1*, *SOCS1*, *SOX2*, *YAP1*, *ZFHX3*Herrgott et al. [116]/2022Differentiate PitNETs from OPD through analysis of LB specimensPitNETs (n = 37)PitNETs release DNA methylation markers in the serum/plasmaSantana-Santos et al. [101]/2022The Northwestern Medicine (NM) classifier of CNS tumors was developed and validated3905 central nervous tumor samples. 2801 samples were used in the original classifier training, and 1104 were used for validation.Whole-genome methylation profiling of brain tumors for clinical testing has been developed and validatedTucker et al. [113]/2023Validate the differential DNA methylation and related MAX protein expression profiles between NFPA and GHPA52 surgically resected tumors (37 NFPA, 15 GHPA)MAX transcription factor binding sites are globally hypomethylated and demonstrate increased accessibility for transcription factor binding in GHPA compared to NFPAGalbraith et al. [102]/2023The clinical utility of DNA methylation for primary diagnosis of brain tumors1921 primary CNS tumorsDNA methylation is of limited diagnostic and prognostic value in the diagnosis of meningioma, schwannoma, and pituitary adenoma Kober et al. [103]/2023Genome-wide DNA methylation patterns in somatotroph tumorsForty-eight tumor samplesDifferences in DNA methylation profiles between three molecular subtypes are undeniableFeng et al. [104]/2023DNA methylation analysis in PPETS tumors and the comparison cohort15 posterior pituitary tumors (PPT)PPETS and PPT form a distinct molecular cluster

#### 2.2.4. Additional Papers

In addition to the three main methods described above, nine other studies explored DNA methylation in pituitary adenomas using alternative approaches. The three case reports used methylation profiling to assist in diagnosing complex tumor cases, distinguishing pituitary adenomas from other intracranial pathologies based on genome-wide methylation patterns [120,121,122]. The other six studies used various mass-array platforms to assess methylation patterns across tumor subgroups and subtypes within pituitary adenomas. Though comparable to MSP, these methods offer technical distinctions that expand analytical scope.

Genes and pathways examined in these studies include the *ZAR1* non-promoter region, noted for frequent methylation events in pituitary adenomas [123]. Additionally, *MGMT* promoter methylation was investigated, with findings indicating significant methylation in carcinomas [124]. Other notable associations included *LINE-1*, linked to tumor invasiveness [125], and *METTL3*, which appears to influence cell growth regulation in GH-secreting pituitary adenomas (GH-PAs) [126]. Two genome-wide studies explored links between methylation status, invasive behavior [127], and proliferation rate [128]. However, no specific genes or pathways were conclusively tied to these characteristics.

Comprehensive findings from these studies are summarized in Appendix A).

Table 2 summarizes the key findings from the studies, using methylation-restricted digestion, methylation-specific PCR, and chip-based DNA methylation analysis. The table summarizes the genes and pathways found to be associated with clinical behavior, those with probable associations, and those showing no association.

## 3. Discussion

For over three decades, DNA methylation has been a valuable tool to distinguish tumor types and subtypes, especially when specific genetic mutations are absent. Epigenetic modifications offer valuable insights into gene regulation and expression, yet defining their direct clinical impact remains a challenge. Early studies using methylation-restricted digestion revealed that the growth hormone gene is methylated in non-functioning pituitary adenomas (NFPAs), but not in functioning pituitary adenomas (FPAs). Since then, our understanding of the role of DNA methylation in pituitary tumor biology has advanced significantly. This review aimed to provide an overview of DNA methylation studies in pituitary adenomas, summarizing the genes and signaling pathways investigated to date and identifying those with potential clinical relevance.

Only a limited number of genes demonstrated a clear correlation between methylation and expression across different analytical platforms. For example, *POMC* promoter methylation was linked to reduced gene expression and was also associated with aggressive features in Cushing’s disease [30,63]. *POMC* plays a crucial role in the synthesis of the adrenocorticotropic hormone (ACTH) in pituitary corticotroph cells, and its dysregulation may impact the hypothalamic–pituitary–adrenal (HPA) axis and potentially influence the onset and severity of corticotropic tumors.

Another gene with notable findings was *SOCS-1*, which was associated with non-functioning pituitary adenomas across multiple modalities, including methylation-specific PCR (MSP) and EPIC array [24,44]. *SOCS-1* (suppressor of cytokine signaling 1) is involved in the JAK-STAT pathway, where it acts as an inhibitor. The dysregulation of this pathway, particularly through *SOCS-1*, has been associated with more aggressive tumor phenotypes [129]. *RASSF1A*, identified both by MSP [41] and EPIC [111], is a well-known tumor suppressor gene that prevents tumorigenesis and uncontrolled cell growth through various intracellular pathways [130]. Methylation of *RASSF1A* would lower expression and increase the risk of tumorigenesis and uncontrolled growth. Additionally, *RB1*, a pivotal tumor suppressor and the first of its kind identified, was detected across multiple analysis methods, although not by EPIC [25,32,38,67,68]. The established role of RB1 in inhibiting invasive growth makes it a key gene of interest in studies aiming to uncover risk factors for aggressive pituitary adenomas [131]. Although multiple studies found a correlation between DNA methylation status for specific genes and sites, the reproducibility of these findings was only seen for the genes mentioned above.

Although the gold standard for methylation analysis remains whole-genome bisulfite sequencing (WGBS), this has yet to be performed on pituitary adenomas. Early studies relying on digestion-restricted analysis provided foundational insights and laid the groundwork for subsequent advances but are now limited by outdated technology. Methylation-specific PCR (MSP) has since offered more targeted assessments, as its primers are designed to focus on relevant gene regions. MSP can yield more precise methylation profiles of genes or pathways and is often paired with gene expression analysis to investigate the functional impact of methylation [132]. The shift toward chip-based methylation analysis appears driven by its accessibility and efficiency, which is particularly advantageous for handling large cohorts of tissue samples [100]. More recent studies have employed both chip-based methylation analysis and expression profiling to examine the correlation between DNA methylation and gene expression [14,98]. These studies provide large exploratory datasets and offer valuable insights into epigenetic regulation. However, they are limited by the scope of the chip itself, which captures less than five percent of the methylation sites across the genome [133]. Although these sites are selected to represent coding regions, the inherent selection bias of commercially available chips can limit conclusions about unmethylated regions and their effects on gene expression. For example, *MGMT* promoter methylation, which has been well-studied using MSP [25,86,87,88,89,90,91,92], has shown no impact on treatment outcomes and has not been consistently identified as a differentially methylated region in chip-based studies.

The clinical applications of DNA methylation in pituitary adenomas have shifted from trying to locate specific sites along the genome responsible for tumorigenesis [42,56], often without clinical follow-up, towards using DNA methylation for profiling for the better prognostication of outcomes [22]. This paper did find some selective probes associated with a higher risk of recurrence. However, these findings were not found using the same method within the same subgroup of pituitary adenomas (non-functioning) [100]. Therefore, the current research trends point towards using DNA methylation for subclassification into subgroups, rather than pinpointing specific methylation sites responsible for tumor regrowth [98]. This is further illustrated by the multiple studies showing a probable association between methylation levels and clinical correlation (Table 2).

This study did encounter some limitations. Multiple studies reported analysis of methylome and transcriptome data [98,100]. Transcriptomic analysis was not included in this paper, and methylation of specific sites did not always correlate with down-regulated expression. Therefore, the inclusion of transcriptomic data and the expression of gene products correlated with next-generation sequencing (NGS), and maybe even proteomics, could probably better illustrate the complex molecular pathways responsible for tumorigenesis, invasive growth, and differences in recurrence rate. In isolation, DNA methylation analysis seems limited when applied for selective genes, as the reproducibility within methods (MSP) for *MGMT* and in-between methods as previously described did not show similar results. However, DNA methylation profiling using multiple methylated sites shows promising results, segregating pituitary adenomas into distinct clusters [14]. Furthermore, a fundamental limitation of search strings in systematic reviews is the potential to miss relevant studies due to narrow scope or rigid terminology [134]. Our search entailed multiple databases to decrease the risk of missing studies [135]. Moreover, we consulted a librarian to optimize the search string to provide a very broad search. Despite optimization, no search string can guarantee full coverage of all pertinent studies.

This review also analyzed the impact of methylation on clinical outcomes, including associations between methylation status, gene expression, and tumor behavior. A variety of study outcomes have been reported, ranging from assessments of gene expression changes linked to methylation in specific genes or promoter regions to broader correlations between methylation profiles and preoperative clinical features, such as invasive growth. Although some studies provide limited postoperative follow-up, more recent investigations are beginning to report recurrence-free survival data in relation to methylation status [108]. These findings suggest that DNA methylation profiling holds promise as a prognostic marker, potentially guiding more tailored postoperative care for patients with pituitary adenomas.

## 4. Materials and Methods

This scoping review followed a protocol registered on medRxiv [136] without deviation and adhered to the Preferred Reporting Items for Systematic Reviews and Meta-Analyses for Scoping Reviews (PRISMA-ScR) guidelines [137]. The initial search block was developed in Embase (Ovid, Elsevier, Amsterdam, The Netherlands) using relevant subject headings and keywords, and was then adapted for use in other bibliographic databases: Medline (Ovid, US National Library of Medicine, Bethesda, MD, USA), Cochrane Central Register of Controlled Trials (CENTRAL, Wiley, Hoboken, NJ, USA), and Scopus (Elsevier, Amsterdam, The Netherlands). Details of the search strategy are provided in the Appendix A).

The inclusion criteria focused on primary studies involving DNA methylation analysis in human pituitary adenomas. Both pre-clinical and clinical studies were included, but review articles were excluded. We included publications in any language, provided that both the title and abstract were in English. Records identified in the database searches were imported into Covidence (Veritas Health Innovation), where duplicates were removed. Covidence facilitated screening at both the title/abstract (Cohens Kappa = 0.69) and full-text (Cohens Kappa = 0.79) levels. Two reviewers (MWM and MJN) independently screened all records, with any disagreements resolved through discussion within the research team. Included records were then exported to Endnote for further data organization.

A data-charting form was developed to systematically extract relevant information from each included study. The following data items were recorded for each study:

Author(s)

Year of publication

Origin/country of origin

Aim/purpose

Population and sample size

Methodology/methods

Intervention type, comparator, and details (e.g., duration, if applicable)

Outcomes and their measurement methods (if applicable)

Key findings related to the review question(s)

## 5. Conclusions

This scoping review explored the role of DNA methylation pituitary adenomas through the last three decades. Epigenetic modifications undeniable contribute to the subclassification, invasive growth and recurrence probabilities of PAs. Specific pathways might be directly correlated with clinical outcomes, in which epigenetic alterations play a major role. Further research into epigenetic mechanisms could enhance the potential of epigenetics as a prognostic marker, enabling personalized follow-up strategies and supporting expanded treatment regimens based on these findings.

## Figures and Tables

**Figure 1 ijms-26-00531-f001:**
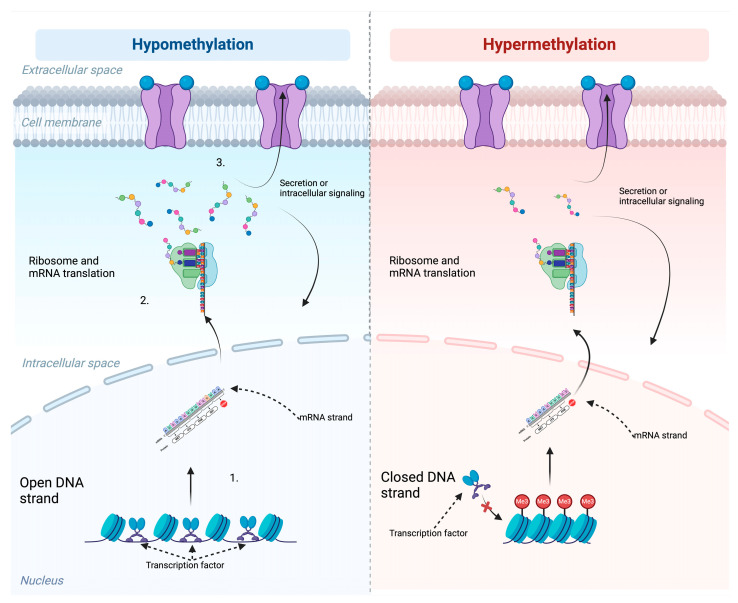
Schematic representation of hypomethylation (**left**) and hypermethylation (**right**) and their effects on gene expression and protein synthesis. 1. Hypomethylation maintains open DNA strands, allowing transcription factors to bind and facilitate gene transcription. In contrast, hypermethylation results in closed DNA conformation, blocking transcription factor binding and suppressing transcription. 2. In hypomethylated regions, mRNA is efficiently transcribed and translated into proteins in ribosomes. Hypermethylation, however, reduces mRNA availability or alters splicing, leading to limited or absent translation. 3. As a result, gene products are fully expressed in hypomethylated states, enabling secretion or intracellular signaling, whereas hypermethylation produces fewer or no gene products, impairing these processes. Created in BioRender. Møller, M.—2025. https://BioRender.com/w17i636 (accessed on 4 January 2025).

**Figure 2 ijms-26-00531-f002:**
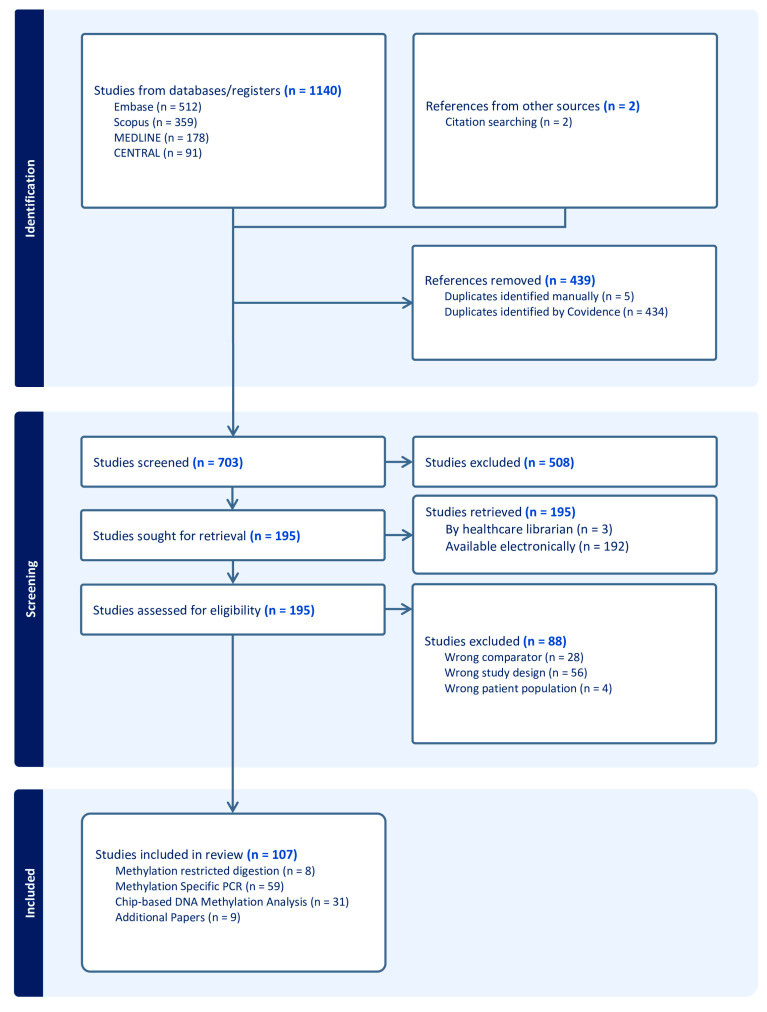
PRISMA flow diagram for identification, screening and included studies on pituitary adenomas and DNA methylation. Numbers in blue font represent the total number of studies at each stage of the process, as indicated within the respective boxes.

**Table 2 ijms-26-00531-t002:** Summary of findings regarding associations between methylation levels and gene expression and clinical outcomes.

Methylation Association	Method	Genes/Pathways	Clinical Outcomes	Key Findings
**Correlation with methylation and expression/clinical behavior**	**Methylation-restricted** **digestion**	*GH-gene*, *p16*, *RB1 promoter*, *POMC promoter*, *1A DMR*	*POMC* promoter methylation repressed expression.Partial methylation of *1A DMR* induced higher expression	No clinical function of these findings
**Methylation-specific PCR**	*CDKN2A*, *DAP kinase*, *GADD45g*, *RASSF1A*, *RASSF3*, *Gal-3*, *SOCS-1*, *FGFR2-IIIb*, *GSTP1*, *NNAT*, *EFEMP1*, *sFRP4*, *p21*, *p27*, *WIF1*, *GIPR*, *DNMT1*, *DNMT3A*, *DNMT3B*, *LAMA2*, *TERT*, *GNAS*, *POMC promoter*	*RASSF1A*, *RASSF3*—somatotroph tumorigensis, *SOCS-1*—NFPAs, *FGFR2-IIIb*—oncogenic signals, *GSTP1*—invasiveness and response to SSA, *sFRP4*—aggresiveness, *p21* and *p27*—invasiveness, *DNMT1*, *DNMT3A,* and *DNMT3B*—oncogenic factors, TERT—disease progression, POMC promoter	Related primarily to invasive behavior
**Chip-based**	*STAT5A*, *RHOD*, *GALNT9*, *RASSF1*, *CDKN1A*, *TP73*, *STAT3*, *HMGA2*, *FAM163A*, *HIF3A*, *PRSS8*	Hypermethylated in NFPA	Different significant sites for NFPA, minimal overlap between studies
*PHYHD1*, *LTBR*, *MYBPHL*, *C22orf42*, *PRR5*, *ANKDD1A*, *RAB13*, *CAMKV*, *KIFC3*, *WNT4*, and *STAT6*	Invasive behavior of NFPA
*FAM90A1*, *ETS2*, *STAT6*, *MYT1L*, *ING2*, and *KCNK1*	Regrowth of NFPA
*C7orf50*, *GNG7*, and *BAHCC1*	Methylated in pituitary adenomas
*DCC*, *DLG5*, *ETS2*, *FOXO1*, *HBP1*, *HMGA2*, *PCGF3*, *PSME4*, *RBPMS*, *RREB1*, *SMAD1*, *SOCS1*, *SOX2*, *YAP1*, *ZFHX3*	Hub proteins for NFPA
Myc-associated protein X transcription factor binding sites	Globally hypomethylated in NFPA
**Other methods**	*METTL3*	Regulation of cell growth or hormone secretion of GH-PA	
**Probable association (no significant findings)**	**Methylation-specific PCR**	*E-cadherin*, *C22orf3*, *MEG3*, *p15INK4b*, *RB1*, *CDH13*, *FGFR2*, *CDKN2C*, *RIZ1*, *SSTR5*, *Pttg1*, autosomal and X-linked genes, *CHST7* promoter	*E-cadherin*—aggressive behavior, *MEG3*—pathogenesis of NFPAs, *CDH13*—pathogenesis, Autosomal and X-linked genes—aggressiveness and response to treatment, *CHST7*—promoter for lineage	Related to pathogenesis
**No association with clinical differences**	**Methylation-restricted** **digestion**	*MGMT* promoter	Poor predictor of outcome of treatment with Temozolomide	No significant association between methylation status of MGMT and clinical outcomes
**Methylation-specific PCR**	*p16*, *CDH1*, *p18INK4C*, *NDRG2*, *MGMT*, *GADD45b*, *CDH1*, *MEN1*, *hTERT* promoter, *STAT3* promoter	*p16*—aggressiveness, *MGMT*—expression, *NDRG2*—invasiveness, *STAT3* promoter—prognostic marker	Genes otherwise proven related to aggressive pathologies showed no association with aggressive behavior in pituitary adenomas
**Chip-based**	*TERT*, *Myc-associated protein X transcription factor binding sites*	*TERT*—pathogenesis,Myc-associated protein X transcription factor binding sites	
**Other methods**	*ZAR1*, *MGMT* promoter, *LINE-1*	*MGMT* promoter—immunoreactivity, *LINE-1*—aggresiveness

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
