# Peer review of "DNA Methylation in Pituitary Adenomas: A Scoping Review"

_ijms, 2025, doi:10.3390/ijms26020531_

Round 1

Reviewer 1 Report

Comments and Suggestions for Authors

Authors have reviewed DNA methylation in pituitary adenomas. Several comments are addressed to the Authors:

1. Introduction - Authors should briefly describe what is exact role of DNA methylation (mDNA) in pituitary neoplasms - impact on survival, treatment outcomes, quality of life, etc.

2. Figure 1 is illegible

3. Table 1 should be reorganized because is voluminious. Maybe width compression  of column 1-3 and extending space of last one could reduce size of that table.

4. Any figure merging mechanisms in which mDNA participate in pituitary tumors will better hel readers to understand this epigenetic mofidication in relation to such group of neoplasms.

5. Literature review sounds too technical because Authors give a particular attention to methods of mDNA analysis, etc. I suppose that for readers more clinical information will be interesting. Please describe how does mDNA of particular genes affect survival or clinical outcomes of patients. Please describe it more carefully. 

6. Some limitations of the study need to be described as well as deficits of mDNA analysis in puitary tumors.

Reviewer 2 Report

Comments and Suggestions for Authors

1. Lines 34–35: It would be helpful to include references to support the observation that classifications based on transcription factors and hormones may not be sufficient. Additionally, the punctuation on line 35 could be revised for clarity.

2. The structure of Section 1.2 could be improved by organizing it into subsections for WGBS, MSP (including methylation-restricted digestion), and ChIP-based DNA methylation analysis. This might enhance the readability and logical flow.

3. The criteria for selecting studies could be explained in more detail to make the review more transparent and informative for readers.

4. For Table 2, it would be better to separate the data on gene expression and clinical outcomes into two tables. Additionally, visual tools, such as Venn diagrams, could help illustrate overlaps between findings from different studies.

5. Including a comparison of DNA methylation methods (e.g., MSP vs. Chip-based analysis) using the same genes, such as the MGMT promoter, might provide valuable insights into the strengths and limitations of each approach.

6. For Table 2, it may be worth discussing why genes like p16 and MGMT appear with different correlations. Providing context about variations between studies or methodologies would help readers better understand these findings.

7. Expanding the discussion to address inconsistencies in findings and potential limitations in current approaches to DNA methylation analysis could strengthen the argument for future research.

8. Ensuring all gene names are italicized would align the manuscript with standard scientific conventions.

9. Focus on Methylation Levels: Consistently emphasizing the relationship between DNA methylation levels and clinical outcomes might help maintain the manuscript’s focus.

Reviewer 3 Report

Comments and Suggestions for Authors

This paper provides a comprehensive review of the literature that exist on DNA methylation in pituitey adenomas detailing new methods and findings. I have only some minor comments:

1)      Figure 1 is a bit large and difficult to interpret , it would be nice to have something simpler.

2)      Line 62-67 : it would be helpful to briefly explain why these methods are chosen over others and their specific advantages.

3)      In line 313 there is a “to” missing in : Epigenetic modifications undenielably contribute the subclassification

Round 2

Reviewer 1 Report

Comments and Suggestions for Authors

The authors have addressed all of my comments. I have no further questions.

Reviewer 2 Report

Comments and Suggestions for Authors

The authors have thoroughly addressed all my concerns. I recommend this manuscript for acceptance and publication.